# A conserved glutathione binding site in poliovirus is a target for antivirals and vaccine stabilisation

Mohammad W. Bahar [1✉], Veronica Nasta[1,2,3], Helen Fox[4], Lee Sherry [5], Keith Grehan [5], Claudine Porta[1,6], Andrew J. Macadam [4], Nicola J. Stonehouse [5], David J. Rowlands [5], Elizabeth E. Fry [1] & David I. Stuart [1,7✉]

Strategies to prevent the recurrence of poliovirus (PV) after eradication may utilise non-infectious, recombinant virus-like particle (VLP) vaccines. Despite clear advantages over inactivated or attenuated virus vaccines, instability of VLPs can compromise their immunogenicity. Glutathione (GSH), an important cellular reducing agent, is a crucial co-factor for the morphogenesis of enteroviruses, including PV. We report cryo-EM structures of GSH bound to PV serotype 3 VLPs showing that it can enhance particle stability. GSH binds the positively charged pocket at the interprotomer interface shown recently to bind GSH in enterovirus F3 and putative antiviral benzene sulphonamide compounds in other enteroviruses. We show, using high-resolution cryo-EM, the binding of a benzene sulphonamide compound with a PV serotype 2 VLP, consistent with antiviral activity through over-stabilizing the interprotomer pocket, preventing the capsid rearrangements necessary for viral infection. Collectively, these results suggest GSH or an analogous tight-binding antiviral offers the potential for stabilizing VLP vaccines.

[1] Division of Structural Biology, University of Oxford, The Henry Wellcome Building for Genomic Medicine, Headington, Oxford OX3 7BN, UK. [2] Magnetic Resonance Center CERM, University of Florence, Via Luigi Sacconi 6, 50019 Sesto Fiorentino Florence, Italy. [3] Department of Chemistry, University of Florence, Via della Lastruccia 3, 50019 Sesto Fiorentino Florence, Italy. [4] The National Institute for Biological Standards and Control, Potters Bar EN6 3QG, UK. [5] Astbury Centre for Structural Molecular Biology, School of Molecular and Cellular Biology, Faculty of Biological Sciences, University of Leeds, Leeds LS2 9JT, UK. [6] The Pirbright Institute, Pirbright, Surrey GU24 0NF, UK. [7] Diamond Light Source, Harwell Science and Innovation Campus, Didcot OX11 0DE, UK. ✉email: mohammad.bahar@strubi.ox.ac.uk; dave.stuart@strubi.ox.ac.uk

The *Enterovirus* (EV) genus of the *Picornaviridae* family contains notable viral pathogens that cause various animal and human diseases and are regarded as potential zoonotic threats[1]. These small non-enveloped RNA viruses contain 60 copies each of the viral proteins VP1-4 arranged to form an icosahedral protein capsid. VP1-3 each comprise a β-barrel with extended surface loops and C-termini that form the outer capsid surface. The N-termini of VP1-3, as well as the whole of VP4 (N-terminal to VP2 in the VP0 precursor), form a network on the inner surface[2]. Several vaccines now exist against enteroviruses, notably against poliovirus (PV) serotypes 1–3 (PV1-3, of species EV-C) and EVA71[3,4]; however, there are no licensed anti-enterovirus drugs. The most promising drug candidates to date have been compounds targeting a hydrophobic pocket internal to the β-barrel of capsid protein VP1[5,6]. These 'capsid binders' or 'pocket factors' mimic a fatty acid moiety found naturally within this pocket, which maintains the native 'D' antigenic conformation. This natural lipid is normally dislodged on receptor engagement, leading to a structural shift involving radial expansion and the appearance of pores in the capsid, forming the 'C' antigenic conformation[7]. By binding more tightly than the natural fatty acid, the action of pocket factors is to block virus-receptor interaction and uncoating[8]. Such antivirals against enteroviruses were discovered many years ago and although one of these, pleconaril, entered clinical trials, it was not taken forward due to possible side effects, and since it was able to inhibit the in vitro replication of PV2 and PV3 but not that of PV1[9].

The World Health Organization polio eradication campaign using inactivated (IPV) and attenuated oral (OPV) polio vaccines has massively reduced the number of poliomyelitis cases, taking the world close to eradication[10]. However, post-eradication, new PV vaccines will be required that are not derived from an infectious virus, such as virus-like particle (VLP) vaccines utilising the capacity for recombinantly expressed capsid proteins to self-assemble into empty viral capsids[11]. Whilst indistinguishable in many respects, these empty capsids are inherently less stable than an infectious virus and prone to switching conformation from the native D-antigenic state to the less immunogenic C-antigenic structure[12]. Specifically, these VLPs lack the stabilising interactions from encapsidated RNA and from the maturation cleavage of VP0, which results in the repositioning of the VP2 N and VP4 C-termini[13] to form an internal scaffold. Stable capsid (SC) mutants have been developed for the VLPs of all three serotypes of PV (PV1 MahSC6b, PV2 MEFSC6b, PV3 SktSC8)[14], and PV3 capsids with 8 mutations are as immunogenic as IPV in the absence of adjuvant. Furthermore, PV VLPs have been successfully produced in recombinant expression systems, establishing a proof-of-principle for their use as potential vaccine candidates for a post-polio eradication era[15–18]. Whilst live virus vaccines remain in use, antivirals may also be required after polio eradication to control outbreaks of circulating vaccine-derived polioviruses[19,20].

Glutathione is an abundant thiol peptide present in animal cells and plays a critical role in maintaining cellular redox potential[21,22]. An imbalance in the ratio between reduced (GSH) and oxidised (GSSG) forms of glutathione is implicated in various enterovirus infections[23] and depletion of cellular GSH levels using the GSH biosynthesis inhibitor L-buthionine sulfoximine (BSO)[24,25] and the small molecule inhibitor TP219[26] blocks the assembly and morphogenesis of many enterovirus capsids. Mutations conferring GSH-independence map to the protomer interface between proteins VP1 and VP3 and are consistent with GSH binding being essential for the formation of pentameric subunits critical to assembly[24,27]. A benzene sulphonamide compound (CP17) and derivatives thereof have been identified, which inhibit viral replication with micromolar affinity in a cell-

based screening assay[28] for a range of EVs, including PV1 and some rhinoviruses. For PV1, the derivative CP48 was more active than CP17. CP17 has been observed bound at a VP1–VP3 interprotomer cavity of Coxsackievirus B3 (CVB3) using cryogenic electron microscopy (cryo-EM)[28,29], the same cavity where GSH has been observed to bind the bovine enterovirus EVF3[30]. This cavity is conserved in many but not all EVs and not generally across other picornaviruses. Furthermore, inhibition of GSH synthesis with BSO in EVF3-infected cells shows GSH dependency for virus growth and dose-dependent stabilisation by GSH of EVF3 virions has been demonstrated by thermal stability assays[30]. This is consistent with earlier studies showing that GSH stabilises certain enteroviruses, including PV during viral morphogenesis[24], as well as being a potent stabiliser of OPV vaccine formulations from heat inactivation[31]. More recently, CP48 has been shown bound to CVB4[29]. Both CP17 and CP48 can act synergistically with pocket factors suggesting they have different antiviral mechanisms that can be exploited to develop anti-enteroviral therapies with improved efficacy and reduced side effects[28,29].

Here we have used PV VLPs as a clinically relevant model system to better understand the GSH-binding site as a druggable pocket for the development of antivirals against EVs and as a site for the potential stabilisation of synthetic vaccine candidates. Using single-particle cryo-EM, we show GSH bound to the VP1–VP3 interprotomer pocket for the stabilised PV3-SC8 VLP. Furthermore, we show that GSH can stabilise VLPs of all serotypes of PV in the native D-antigenic conformation required for eliciting an immune response. In addition, a high-resolution cryo-EM analysis (at better than 2.0 Å resolution) is presented for CP17 bound to a VLP of wild-type (wt) PV2. It was, however, not possible to observe binding of CP48 using similar protocols. The interactions between CP17 and the ligand binding site are analysed and compared to those seen for CP17 and CP48 in CVB3 and CVB4, respectively, allowing key conserved residues to be identified. Sequence differences in the VP1–VP3 interprotomer cavity between PV and other EVs may explain altered binding orientations and specificities of the benzene sulphonamide derivatives observed in the wt PV2-CP17 complex studied here, compared to other EVs such as CVB3 and CVB4. Collectively, these results further increase understanding of the biological role of the VP1–VP3 interprotomer pocket, as well as that of a new potential antiviral drug class against EVs. Exploiting these findings opens up strategies for GSH or synthetic analogues thereof to be used for targeting the druggable pocket to stabilise PV VLPs, which may be important for next-generation vaccines.

## Results

**GSH binds the VP1–VP3 interprotomer pocket on the surface of the PV3-SC8 VLP.** To investigate the structural basis of whether and how GSH binds to PV capsids, we chose to examine the complex of GSH with a previously characterised PV VLP, PV3-SC8[14]. Since PV3-SC8 has previously been shown to bind pocket factors such as GPP3 in the VP1 hydrophobic pocket[18], we investigated in situ if GSH binding interferes with this stabilisation mechanism by incubating a molar excess of GPP3 (VLP:compound molar ratio of 1:300) followed by GSH at 10 mM to form a ternary complex (PV3-SC8$^{GPP3+GSH}$) with the PV3-SC8 VLP expressed in yeast[16]. This complex was applied to EM grids that were rapidly vitrified for single-particle cryo-EM data collection. Image processing of a final set of 5364 particles yielded a 2.5 Å icosahedral reconstruction as assessed using the FSC 0.143 threshold criterion[32] (Supplementary Fig. 1 and Tables 1, 2). The cryo-EM electron potential map revealed the VP0, VP1 and VP3 capsid protein subunits of the PV3-SC8 VLP to be well ordered

**Table 1 Cryo-EM data collection and image processing.**

| | PV3-SC8[GPP3+GSH] (EMD-15725) (PDB 8AYX) | PV3-SC8[pleconaril+GSH] (EMD-15726) (PDB 8AYY) | wt PV2-CP17 (EMD-15727) (PDB 8AYZ) |
|---|---|---|---|
| **Data Collection** | | | |
| Voltage (kV) | 300 | 300 | 300 |
| Magnification (×) | 129629 | 129629 | 60314 |
| Defocus range (µm) | −2.9 to −0.8 | −2.9 to −0.8 | −2.3 to −0.8 |
| Dose rate (e$^-$/pixel/s) | 0.59 | 35.79 | 14.02 |
| Frames | 60 | 25 | 50 |
| Frame length (s) | 1.163 | 0.046 | 0.034 |
| Total electron dose (e$^-$/Å$^2$) | 34.89 | 35.29 | 34.68 |
| Micrographs | 2503 | 4706 | 4750 |
| **Data processing** | | | |
| Pixel size (Å) | 1.08 | 1.08 | 0.829 |
| Initial particles (no.) | 19622 | 34523 | 157002 |
| Final particles (no.) | 5364 | 15275 | 51518 |
| Box size (pixels) | 450 | 450 | 450 |
| Symmetry | I1 | I1 | I1 |
| Accuracy of rotations (°) | 0.1325 | 0.1845 | 0.1085 |
| Accuracy of translations (Å) | 0.2160 | 0.3175 | 0.1658 |
| Resolution (Å) | 2.54 | 2.64 | 1.88 |
| Map sharpening $B$-factor (Å$^2$) | −52.3 | −72.2 | −15.0 |

**Table 2 Structure refinement and validation for the capsid protein (VP0, VP1 and VP3).**

| | PV3-SC8[GPP3+GSH] (EMD-15725) (PDB 8AYX) | PV3-SC8[pleconaril+GSH] (EMD-15726) (PDB 8AYY) | wt PV2-CP17 (EMD-15727) (PDB 8AYZ) |
|---|---|---|---|
| **Model composition** | | | |
| Non-hydrogen atoms | 5863 | 5857 | 6565 |
| Protein residues | 736 | 736 | 802 |
| Ligands | GPP3:1 | Pleconaril:1 | Sphingosine:1 |
| | GSH:1 | GSH:1 | CP17:1 |
| Waters | | | 262 |
| **Refinement** | | | |
| Resolution (Å) | 2.54 | 2.64 | 1.88 |
| Map CC[a] (Mask) | 0.85 | 0.87 | 0.90 |
| Map CC[a] (Volume) | 0.83 | 0.85 | 0.88 |
| **RMS deviations** | | | |
| Bond lengths (Å) | 0.002 | 0.003 | 0.003 |
| Bond angles (°) | 0.470 | 0.455 | 0.549 |
| **Mean B-factor (Å$^2$)** | | | |
| Protein | 20.36 | 23.74 | 24.16 |
| Ligand | 19.37 | 23.69 | 24.60 |
| Water | | | 23.26 |
| **Validation** | | | |
| Molprobity[b] score (percentile) | 1.02 (100th) | 0.91 (100th) | 1.11 (100th) |
| Clashscore[b], all atoms (percentile) | 2.42 (99th) | 1.65 (99th) | 2.42 (99th) |
| Ramachandran favoured (%) | 98.20 | 98.20 | 97.60 |
| Ramachandran allowed (%) | 1.80 | 1.80 | 2.40 |
| Ramachandran outliers (%) | 0.00 | 0.00 | 0.00 |
| Rotamer favoured (outliers) (%) | 98.29 (0.16) | 98.13 (0.31) | 98.69 (0.15) |
| Cβ deviations >0.25 Å (%) | 0.00 | 0.00 | 0.00 |
| CaBLAM outliers (%) | 1.12 | 1.26 | 1.28 |
| CA Geometry outliers (%) | 0.56 | 0.28 | 0.26 |
| **EMRinger[c] score** | 4.99 | 4.90 | 7.84 |

[a]Map CC is given for the full particle reconstruction.
[b]Williams et al. (2018) Protein Sci 27:293–315[51].
[c]Barad et al. (2015) Nature Methods 12:943–946[52].

and there was an unambiguous feature for bound GSH in the interprotomer pocket formed between VP1 subunits from two adjacent protomers and VP3 from a single protomer (Fig. 1a–e); the same surface pocket shown to bind GSH in bovine enterovirus EVF3[30]. A structure of the unbound native antigenic form of the same yeast-expressed VLP (PDB ID 8ANW[33]) facilitated comparisons. The overall structure of the GSH-bound PV3-SC8 VLP (Fig. 1a, b, d) is essentially identical to the unbound native antigenic form of PV3-SC8 VLP expressed previously in the plant, mammalian and yeast cells[15,18,33], with root-mean-square deviation (RMSD) in Cα atoms of 0.68, 1.39 and 0.77 Å, respectively (Supplementary Fig. 2a). As expected GPP3 was

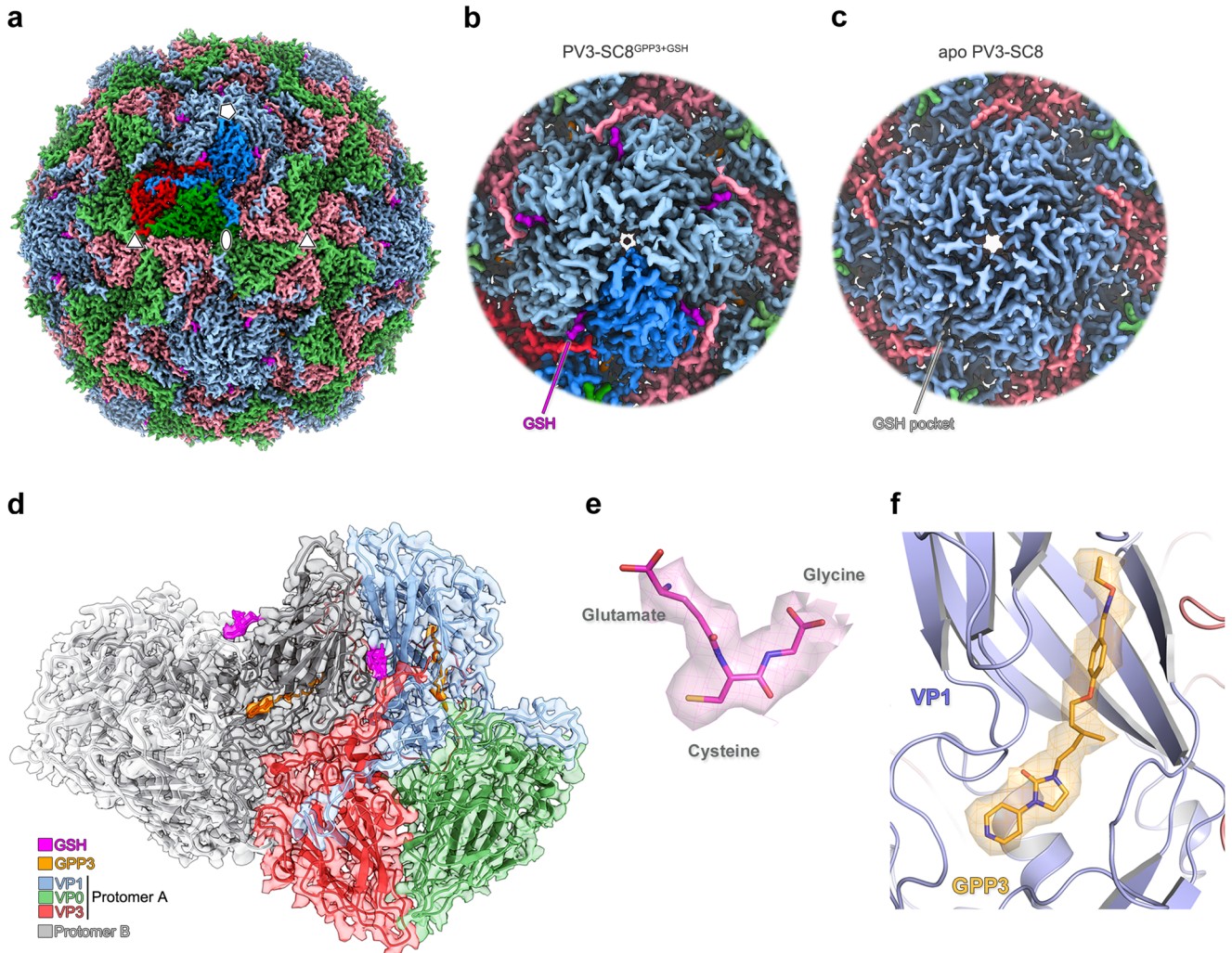

**Fig. 1 Cryo-EM structure of the PV3-SC8<sup>GPP3+GSH</sup> complex. a** Three-dimensional reconstruction of PV3-SC8 VLP after incubation with a molar excess of GSH and GPP3. The VLP is viewed along the icosahedral twofold symmetry axis with the VP1, VP0 and VP3 subunits of the capsid protomer coloured light blue, light green and light red, respectively. A single capsid protomer is coloured in a darker shade, GSH magenta and GPP3 orange. **b** Close-up view looking down the icosahedral fivefold symmetry axis, coloured as in **a** showing GSH bound at an interprotomer interface. **c** Close-up view along the fivefold axis of the apo PV3-SC8 VLP reconstruction (EMD-15543) showing the absence of features for GSH. **d** Cartoon representation of two neighbouring capsid protomers; protomer A is coloured as in **a**, protomer B is coloured grey, GSH magenta and GPP3 orange. The cryo-EM map is depicted as a semi-transparent surface. **e** Cryo-EM electron potential map with GSH fitted as a stick model and elemental colouring for oxygen (red), nitrogen (blue) and sulfur (yellow). The map is displayed at a contour level of 1.2 σ (σ is the standard deviation of the electron potential map). **f** Cartoon depiction of the VP1 hydrophobic pocket with bound GPP3 fitted into the cryo-EM potential map. The electron potential map for GPP3 is shown at 1.5 σ. All electron potential map images are rendered at a radius of 2 Å around atoms.

observed bound within the hydrophobic pocket of VP1, in an essentially identical conformation to that previously observed for this compound bound to plant expressed PV3-SC8[18] (Fig. 1d, f and Supplementary Fig. 2b, c). This confirmed that the binding of GSH did not hinder the ability of the VP1 hydrophobic pocket to accommodate capsid-binding drugs such as GPP3, so effects at the two druggable sites, separated by some 22 Å in the capsid are likely to be additive. In the apo structure of PV3-SC8 (PDB ID 8ANW), no features were observed for GSH at the interprotomer pocket (Fig. 1c), or for GPP3 in the VP1 hydrophobic pocket, with the latter instead occupied by the naturally acquired lipid from the yeast cell, modelled as sphingosine based on the length of the carbon chain fitting the cryo-EM potential map[33]. The lack of GSH in the apo structure most likely reflects a relatively fast off-rate, leading to its loss during extensive purification. We also confirmed these results in a parallel experiment by determining a 2.6 Å cryo-EM reconstruction of PV3-SC8 bound to GSH and an

alternative VP1 hydrophobic pocket binding drug, pleconaril (Supplementary Figs. 1, 2d–g). The PV3-SC8<sup>GPP3+GSH</sup> and PV3-SC8<sup>pleconaril+GSH</sup> complexes are essentially identical in backbone conformation for the VP1, VP0 and VP3 subunits of the capsid protomer and binding mode of GSH, with a Cα RMSD of 0.19 Å between GPP3 + GSH and pleconaril + GSH forms of the complex (Supplementary Fig. 2a).

The GSH molecule observed bound in the VP1–VP3 surface pocket of the PV3-SC8 VLP is anchored in place by hydrogen bonds, salt-bridges and hydrophobic interactions (Fig. 2a, b). The carboxyl group of the GSH glycine moiety forms hydrogen bonds with two critical arginine residues; Arg257 and Arg242, contributed from neighbouring VP1 subunits of two capsid protomers (Figs. 1d, 2a, b). Additional hydrogen bonds are formed between Asp169 and Gln174 of VP1 with the amide and carbonyl oxygen groups of the cysteine moiety of GSH, respectively (Fig. 2a, b).

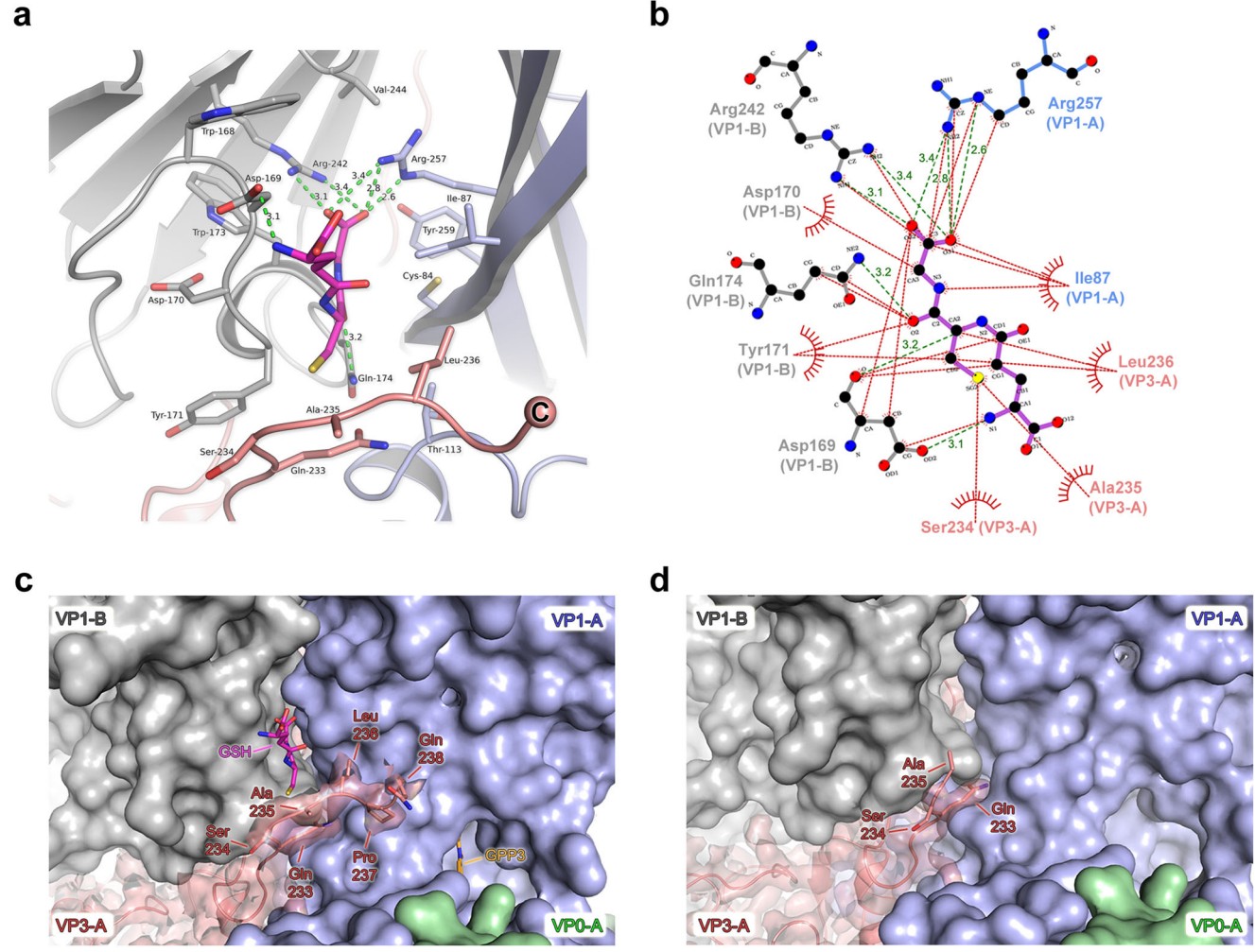

**Fig. 2 Glutathione interactions in the VP1–VP3 interprotomer pocket of the PV3-SC8 VLP and stabilisation of the VP3 C-terminus. a** Cartoon representation of GSH (magenta stick model) bound in the interprotomer surface pocket between neighbouring capsid protomers of the PV3-SC8 VLP. VP1 and VP3 of protomer A are coloured light blue and light red, respectively, and VP1 of protomer B is coloured grey. Amino acid side chains for residues of VP1 and VP3 forming the GSH-binding pocket are shown as sticks and labelled. Hydrogen bond and salt–bridge interactions are shown as green dashes and distances labelled. **b** Ligplot+[61] representation of the GSH-binding pocket showing details of key interactions. Hydrogen bonds and salt-bridges are shown as green dashed lines and hydrophobic interactions as red arcs. **c** Cryo-EM electron potential map of the VP3 C-terminus (red) in the PV3-SC8[GPP3+GSH] complex showing the final three residues are ordered upon GSH binding compared to **d** the apo structure of PV3-SC8 without GSH bound (PDB ID 8ANW, EMD-15543). The electron potential map for VP3 is shown at 1.4 σ and rendered at 2 Å around atoms. VP1 subunits of protomer A and B are shown as molecular surfaces and coloured as in **a** and GPP3 bound in the VP1 pocket is shown as an orange stick model.

**GSH stabilises the C-terminus of VP3 to form the binding pocket**. In apo PV3-SC8 (Fig. 1c) and other native PVs[34], the final three residues of the VP3 C-terminus (Leu236, Pro237 and Gln238) are disordered. In the PV3-SC8[GPP3+GSH] and PV3-SC8[pleconaril+GSH] complexes, we observed that GSH binding resulted in some or all of these residues becoming ordered in the cryo-EM potential map, with Leu236 forming part of the binding site (Fig. 2c, d), so that hydrophobic interactions from Ser234, Ala235 and Leu236 of VP3 form a cap stabilising GSH in the interprotomer surface pocket (Fig. 2b, c). For PV3-SC8[GPP3+GSH] clear structure was observed in the cryo-EM map up to the end of the C-terminus (Gln238), whereas for PV3-SC8[pleconaril+GSH] only an additional two residues (Leu236 and Pro237) became ordered upon GSH binding. For the apo form of PV3-SC8 with no GSH bound (PDB ID 8ANW), no features were observed in the cryo-EM map beyond Ala235 (Fig. 2d). In total, 389 Å$^2$ of solvent-accessible surface area for GSH is buried in the inter-protomer surface pocket, representing ~78 % of the total solvent-accessible surface area of the molecule.

**The GSH binding site is highly conserved in enteroviruses**. The key interacting residues in the GSH-binding site are conserved across a panel of representative enteroviruses, including the three serotypes of PV and EVF3[30] (Fig. 3a, b). Notably, in VP1, Tyr259, Arg242, Arg257, Trp173 and Gln174 are strictly conserved and Asp169 is highly conserved (Fig. 3a). These residues form the core of the binding site and stabilise the bound GSH in the interprotomer pocket; with Arg242 and Arg257 of VP1 from neighbouring protomers in the interface conferring strong positive charge characteristics to the GSH-binding pocket and forming hydrogen bond and salt–bridge contacts with the carboxy terminus of GSH (Fig. 2b). This positively charged patch is a conserved feature in other EVs as seen in the EVF3 GSH complex[30] (Fig. 3c, d). The Leu236 residue that forms hydrophobic interactions from the stabilised VP3 C-terminus with bound GSH in the pocket is chemically conserved across EVs along with other VP3 C-terminal residues (Fig. 3b). Figure 3e shows a structure-based phylogenetic tree of representative enteroviruses coloured according to the similarity of the GSH

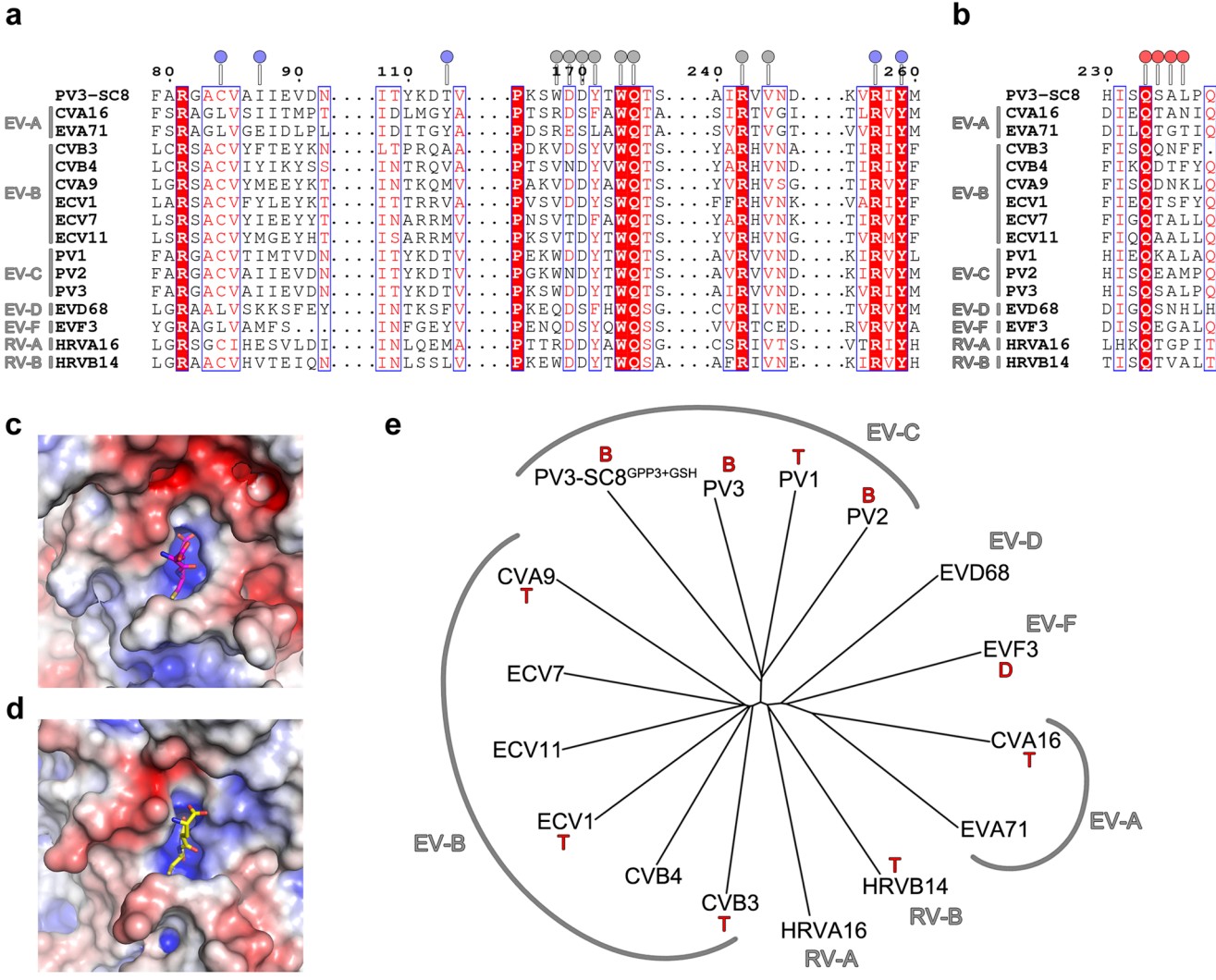

**Fig. 3 Conservation of GSH-binding pocket in enteroviruses. a** Multiple sequence alignment of the VP1 and **b** VP3 subunit sequences of PV3-SC8 with several major enteroviruses. Amino acid residues that form the GSH-binding site are marked with circles above the sequence and coloured blue (VP1 protomer A), grey (VP1 protomer B) and red (VP3 protomer A). Enterovirus species A, B, C, D, F and rhinovirus A and B are marked alongside the alignments. **c, d** Electrostatic charges (±5 kT e⁻¹) mapped onto the molecular surface of **c** PV3-SC8$^{GPP3+GSH}$ and **d** the EVF3 GSH complex and coloured from blue (positive charge) to red (negative charge). **e** Phylogenetic tree derived from a structure-based alignment of the representative set of major enteroviruses aligned in **a**. Those reported to be dependent/stabilised by GSH are marked: T where this was reported in reference [26], D where reported in reference [30] and B where reported in this paper. Abbreviations for each branch of the tree (PDB ID): CVA16 coxsackievirus A16 (5C4W), EVA71 enterovirus A71 (3VBH), CVB3 coxsackievirus B3 (6ZCL), CVB4 coxsackievirus B4 (6ZCK), CVA9 coxsackievirus A9 (1D4M), ECV1 echovirus 1 (1EV1), ECV7 echovirus 7 (2 × 51), ECV11 echovirus 11 (1H8T), PV1 poliovirus type 1 Mahoney (1HXS), PV2 poliovirus type 2 Lansing (1EAH), PV3 poliovirus type 3 Sabin (1PVC), PV3-SC8$^{GPP3+GSH}$ poliovirus type 3 Saukett in complex with GPP3 and GSH (8AYX), EVD68 enterovirus D68 (4WM8), EVF3 enterovirus F3 (6T4C), HRVA16 human rhinovirus 16 (1AYM), HRVB14 human rhinovirus 16 (4RHV).

contacting residues. The viruses labelled on this figure are reported to be dependent on, or stabilised by GSH, and cover all the major branches of the phylogenetic tree, however, there is a suggestion that the situation in some viruses, for example, EVA71, might be more complex[23].

**GSH has a stabilising effect on PV VLPs in vitro.** The role of GSH in facilitating virus assembly for enteroviruses like Coxsackieviruses and PV is well established[26], and the structural basis for this is now understood[30]. To investigate the effect on the physico-chemical properties of VLPs rather than virus particles, the antigenic state of PV VLPs was investigated following heating in the presence of various concentrations of GSH (Fig. 4 and Table 3). Both yeast and mammalian (hamster, BHK-21) cells

expressed PV VLPs were examined for selected examples of the three serotypes with and without stabilising mutations. GSH markedly increased the stability of the mammalian-expressed serotype 3 and serotype 2 VLPs (Fig. 4 and Table 3) and the serotype 1 and 2 yeast VLPs so that they became considerably more stable than IPV (Table 3), although surprisingly, there was little effect on the stability of the yeast-expressed PV3-SC8 particles (Table 3). This was unlikely to be because particles from yeast were already associated with GSH as the cryo-EM structure of purified yeast-derived PV3-SC8 particles showed that there was no specific chemical entity bound at the GSH-binding site (Fig. 1c). It may possibly reflect differences in the occupancy of the hydrophobic pocket in VP1, or at another site affecting stability, since the ternary complexes of PV3-SC8$^{GPP3+GSH}$ and PV3-SC8$^{pleconaril+GSH}$ demonstrate that the bound factors could

act additively or synergistically. The properties of the yeast-expressed PV3-SC8 particles were exceptional as GSH was found to stabilise all other particles tested irrespective of serotype, expression platform or whether stabilising mutations had been introduced (Table 3). Nearly all the stabilisation effect was observed at a GSH concentration of 1 mM, suggesting that this might be a practicable additive for vaccine formulation.

**Other enterovirus inhibitors bind PV VLPs at the same interprotomer pocket as GSH.** Recent work has shown a class of benzene sulphonamide compounds bound to enteroviruses[28,29] at the same interprotomer surface pocket as GSH[30]. We investigated whether these compounds might also bind PV VLPs. Since GSH was shown to effectively stabilise wt PV2 VLPs (Table 3) they were soaked with a molar excess of the benzene sulphonamide derivative CP17 (VLP:CP17 molar ratio of 1:2500), applied to grids and vitrified for single-particle cryo-EM. After image processing, a final set of 51,518 particles yielded a 1.97 Å icosahedral reconstruction for the wt PV2-CP17 complex (FSC 0.143, Supplementary Fig. 3a, b). Ewald sphere correction[35], yielded only a small improvement to 1.88 Å (see Methods, Table 1 and Supplementary Fig. 3a). The resultant electron potential maps revealed highly detailed features for the backbone and sidechain conformations of the VP0, VP1 and VP3 subunits of the capsid protomer (Fig. 5a, b and Supplementary Fig. 3c). Initial maps sharpened with the automatically estimated $B$-factor of $-53.6\,\text{Å}^2$ revealed a distinctive 'L-shaped' feature bound at the same interprotomer pocket occupied by GSH in the ternary complexes of PV3-SC8$^{\text{GPP3+GSH}}$ and PV3-SC8$^{\text{pleconaril+GSH}}$, as well as the

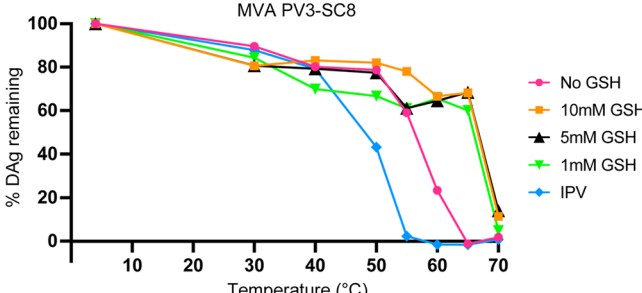

**Fig. 4 GSH has a stabilising effect on PV VLPs in vitro.** Proportion of D antigen reactivity remaining after heating in the presence or absence of 1, 5 and 10 mM GSH. Aliquots of IPV and VLPs were incubated at a range of temperatures and analysed by D Antigen ELISA; reactivity is expressed relative to samples incubated at 4 °C for the same period. Data were shown for PV3-SC8 VLPs expressed in mammalian cells.

EVF3 GSH complex[30]. Maps sharpened less aggressively ($B$-factor of $-15.0\,\text{Å}^2$) confirmed unambiguously that this feature was due to the presence of CP17 bound at the interprotomer pocket of the wt PV2 VLP (Fig. 5a–c).

Like the binding mode for GSH, CP17 is bound within the VP1–VP3 interprotomer pocket of the wt PV2 VLP through two critical arginine residues—Arg243 and Arg258—from neighbouring VP1 subunits, which form hydrogen bonds and salt-bridges with the carboxylate group of CP17 (Fig. 5d, e). The bulk of CP17 forms hydrophobic interactions with the network of residues forming the VP1–VP3 interprotomer pocket of the wt PV2 VLP (Fig. 5e). The wt PV2-CP17 structure is similar to that of the CVB3-CP17 complex (RMSD in Cα atoms of 1.1 Å)[28,29], but there are differences in the precise orientation of CP17 as bound to wt PV2 VLP and CVB3 virus (Fig. 5f), although the key charge interactions with the carboxylate group of CP17 are maintained in both cases. In the CVB3-CP17 structure, CP17 sits upright in the interprotomer pocket due to stacking interactions against Phe76 of VP1 and Phe236 of VP3 from CVB3[29]. In the wt PV2-CP17 complex, the compound lies flat due to sequence differences between PV2 and CVB3, where VP1-Phe76 in CVB3 is replaced by VP1-Ile89 in wt PV2 (Fig. 5f). The interaction of the C-terminal residue VP3-Phe236 in CVB3 is absent in the wt PV2-CP17 structure as the C-terminus of VP3 is disordered in the latter, the final observed residue in wt PV2 being VP3 Ala235 (Fig. 5f). This enables CP17 to adopt a different conformation. The presence of Ala88 in VP1 of wt PV2 compared with Tyr75 at this position in the CVB3-CP17 structure avoids a steric clash that would otherwise disrupt the orientation of the compound in the wt PV2-CP17 complex (Fig. 5f). These sequence differences result in a change of ~52 ° in orientation between CP17 bound to CVB3 and wt PV2 (Fig. 5f). In total 520 Å² of solvent-accessible surface area for CP17 is buried in the interprotomer surface pocket of the wt PV2 VLP, representing ~87% of the total solvent-accessible surface area of the molecule.

Overall, GSH and CP17 occupy similar space within the VP1–VP3 interprotomer pockets of PV3-SC8 and wt PV2 VLPs, respectively, with substantial overlap between the volumes of the two compounds. In addition, the key points of interaction are broadly conserved, notably, the salt–bridge interactions between carboxylate groups on the ligands and arginine residues from adjacent VP1 subunits, stabilising the pentameric association of the protomers, as expected from the role of GSH in the assembly of the pentamers[24], whilst elsewhere there are hydrophobic interactions, more pronounced for CP17 (Fig. 5d). Interestingly, the greater bulk of CP17 means that the C-terminal residues of VP3, which becomes ordered on GSH binding remains disordered upon binding of CP17. An opportunity for increased affinity might therefore come from utilising specific interactions

**Table 3 Effect of GSH on PV VLP thermostability.**

| VLP Prep* | Temperature (°C) at which 50% D-Antigenicity was lost (Temperature difference between treated and non-treated samples) | | | |
|---|---|---|---|---|
| | No GSH | +1 mM GSH | +5 mM GSH | +10 mM GSH |
| IPV (type 3) | 50.0 °C | | | |
| MVA PV2 MEFSC6b | 52.0 °C | 55.0 °C (↑3 °C) | >55.0 °C (↑>3 °C) | >70.0 °C (↑>18 °C) |
| MVA PV3 SktSC8 | 56.0 °C | 66.0 °C (↑10 °C) | 67.0 °C (↑11 °C) | 67.0 °C (↑11 °C) |
| MVA wt PV2 | 34.5 °C | 40.5 °C (↑6 °C) | 41.5 °C (↑7 °C) | 42.5 °C (↑8 °C) |
| MVA wt PV3 | 35.0 °C | | 41.0 °C (↑6 °C) | |
| Yeast PV1 MahSC6b | 43.5 °C | 48.0 °C (↑4.5 °C) | 52.5 °C (↑9 °C) | 51.5 °C (↑8 °C) |
| Yeast PV2 MEFSC6b | 45.5 °C | 48.0 °C (↑2.5 °C) | 50.0 °C (↑4.5 °C) | 50.0 °C (↑4.5 °C) |
| Yeast PV3 SktSC8 | 55.0 °C | 55.5 °C (↑0.5 °C) | 53.0 °C (↓2 °C) | 55.0 °C (=) |

IPV inactivated polio vaccine (derived from virus).
*Both mammalian-expressed (MVA) and yeast-expressed VLPs had naturally derived pocket factors.

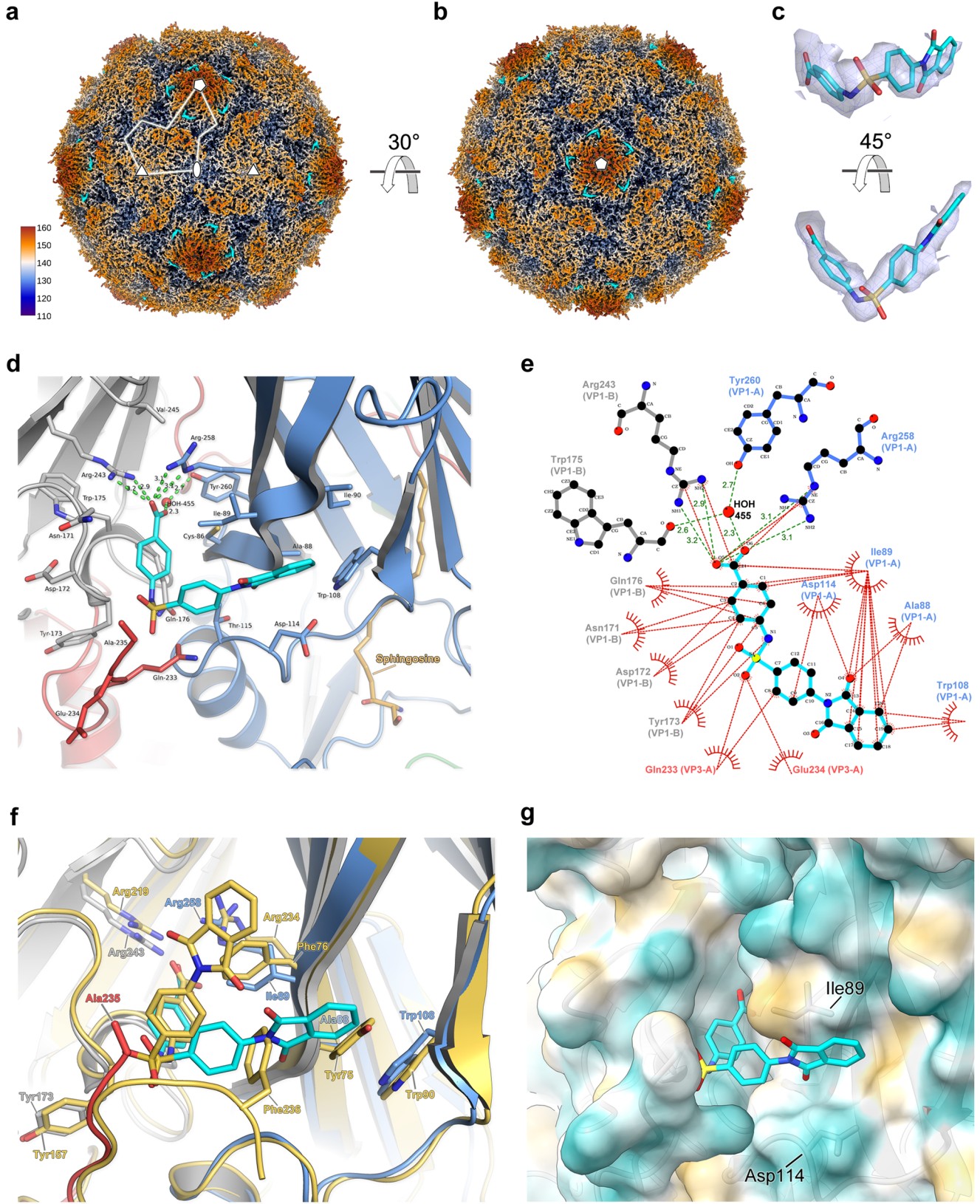

with these terminal residues, although there would be an entropic penalty to their ordering.

It is interesting to note that CP17, which was originally designed to target the lipid-binding pocket within VP1, binds very specifically at a completely separate site and shows no evidence of binding

within the targeted pocket; the features there being entirely consistent with the binding of natural lipids in the wt PV2-CP17 structure (Supplementary Fig. 3d) or that of the pocket factors in the PV3-SC8$^{GPP3+GSH}$ and PV3-SC8$^{pleconaril+GSH}$ complexes (Fig. 1f and Supplementary Fig. 2f). We note that the binding of CP17 is

**Fig. 5 Cryo-EM structure of the wt PV2-CP17 complex. a** Three-dimensional reconstruction of wt PV2 VLP after incubation with a molar excess of the benzene sulphonamide CP17. The VLP is viewed along the icosahedral twofold symmetry axis and coloured by radial distance in Å from the particle centre according to the colour key provided. Cryo-EM electron potential features for CP17 are coloured cyan. A single kite-shaped capsid protomer is outlined in white and five, three and twofold symmetry axes labelled with symbols. **b** View looking down the icosahedral fivefold symmetry axis, coloured as in **a** showing CP17 bound at the interprotomer interface. **c** CP17 shown in close-up as a cyan stick model fitted into the cryo-EM electron potential map, with elemental colouring for oxygen (red), nitrogen (blue) and sulfur (yellow). The map is displayed at a contour level of 1.0 σ and rendered at a radius of 2 Å around atoms. **d** Cartoon representation of CP17 (cyan stick model) bound in the VP1–VP3 interprotomer surface pocket between neighbouring capsid protomers. VP1 and VP3 of protomer A are coloured blue and red, respectively, and VP1 of protomer B is coloured grey. Amino acid side chains for residues of VP1 and VP3 forming the CP17 binding pocket are shown as sticks and labelled. Hydrogen bond and salt–bridge interactions are shown as green dashes and distances labelled. The sphingosine lipid modelled into the VP1 subunit hydrophobic pocket is shown as an orange stick model. **e** Ligplot+[61] representation of the CP17 binding pocket showing details of key interactions. Hydrogen bonds and salt-bridges are shown as green dashes with distances labelled, and hydrophobic interactions as red arcs. **f** Superposition of the CP17 binding sites between wt PV2 VLP and CVB3[29]. VP1, VP3 and CP17 of wt PV2-CP17 are coloured as in **d**, and the CVB3 pocket and CP17 are coloured pale gold. Side chains interacting with CP17 from wt PV2 and CVB3 are labelled and colour coded. **g** Semi-transparent surface representation of the wt PV2-CP17 pocket coloured by hydrophobicity from dark cyan (hydrophilic) to dark gold (hydrophobic). CP17 is shown as cyan sticks and residues of note (see Discussion) are labelled.

reported to be weak (>400 μM) against PV1[28] and not therefore suitable for use directly as a therapeutic candidate or even for vaccine formulation.

CP48 is reported to be a tighter PV binder than CP17[28] and has been observed bound to CVB4[29]. We, therefore, attempted to also determine the structure of the complex of CP48 with wt PV2. However, analysis at 2.7 Å resolution provided no evidence of binding (data not shown). We repeated the analysis with a compound derived directly from the research group that had shown binding to CVB4; however, there was no change in the result, suggesting that this molecule does not bind PV2, despite the reported $EC_{50}$ of 27 nM against PV1[28].

## Discussion

The data presented here, together with previous publications[29,30], show that a GSH-binding site is conserved across many enteroviruses. The fact that these are spread across a structure-based phylogenetic tree of enteroviruses supports the hypothesis that there is a unified assembly mechanism common to the majority of enteroviruses (Fig. 3e). The binding of a single GSH molecule locks together two adjacent VP1 subunits within a pentamer and in the process confers order on the C-terminus of the VP3 subunit, providing a mechanism of stabilisation by GSH during capsid morphogenesis and the formation of pentameric assemblies. It should be noted that the GSH-binding pocket is not conserved in the expanded C-antigenic poliovirus structure[36]. In the latter, the pocket is opened by approximately 5 Å so that the key stabilising interactions cannot be made (Supplementary Fig. 4), explaining the observed stabilisation of the D-antigenic state by GSH. GSH, therefore, stabilises only the native D-antigenic state and D→C conversion would be expected to eject both GSH and the pocket factor. Similar conformational changes between two particle states are seen in other GSH-binding enteroviruses, and thus the use of GSH affinity columns for enterovirus purification will likely select for the D-antigenic states and could lead to loss/removal of material for less stable viruses and VLPs[37].

In line with previous results[28,29], putative antiviral CP17 also binds in the GSH pocket in poliovirus, suggesting that, since it has a conserved biological function, this site may offer the potential for broad-spectrum inhibition of enteroviruses. Surprisingly, however, we were unable to observe the binding of CP48, despite this compound being reported as having a higher affinity for PV (the reported potencies of CP17 and CP48 against PV1 Mahoney are >400 and 27 μM $EC_{50}$, respectively)[28]. Nevertheless, the high-resolution data we provide for both GSH and CP17 binding inform the pharmacophore definition and optimisation of small molecules to bind more strongly at the GSH

pocket than GSH or CP17. Specifically, we suggest that the benzyl portion of the isobenzopyrrole-1,3-dione moiety of CP17 might be modified to provide groups able to make polar or charge interactions with the carbonyl oxygen of VP1-Ile89 or the carboxylate of the sidechain of VP1 Asp114 (Fig. 5e, g).

In addition to the potential of such compounds as antivirals they (and indeed GSH) may prove useful for the stabilisation of vaccine candidates against enteroviruses, namely VLP-based vaccines such as those we are developing to provide poliovirus vaccines for the post-eradication era[15,16,18]. Finally, since the mechanism of action of the GSH-pocket binders and the VP1 lipid-pocket binders is quite separate, they should act additively as either inhibitors or stabilisers, with reduced risk of resistance mutations. It may be interesting to experimentally address such potential synergistic effects, which could be exploited to provide a pathway to the development of new therapeutics against a range of disease-causing enteroviruses.

## Methods

**Production of PV VLPs using yeast and mammalian expression systems**. The production and purification of PV VLPs using yeast and mammalian expression systems has been described elsewhere[15,16].

For yeast-expressed PV VLPs, the P1 gene of PV3-SC8 and an uncleavable 3CD gene also derived from PV3 were codons optimised for expression in *Pichia pastoris*. Both genes were cloned separately into the pPink-HC expression vector multiple cloning site (MCS) using *EcoRI* and *FseI* (NEB). Subsequently, a dual promoter expression vector was constructed through PCR amplification from position 1 of the 3CD pPink-HC to position 1285, inserting a *SacII* restriction site at both the 5′ and 3′ end of the product. The P1 expression plasmid was linearised by *SacII* (NEB), and then the 3CD PCR product inserted. The resulting plasmid was linearised by *AflII* digestion (NEB) and transformed into PichiaPink™ Strain one (Invitrogen, USA) by electroporation.

Transformed yeast cells were screened for high-expression clones by small-scale expression experiments (5 ml cultures), with levels for each clone determined by immunoblotting. For VLP production, cultures were brought to high density in 200 ml YPD in 2 L baffled flasks. After 24 h, the cells were pelleted at 1500×*g* and resuspended in YPM (methanol 0.5% v/v) to induce protein expression and cultured for a further 48 h at 28 °C. Cultures were fed an additional 0.5% v/v methanol at 24 h post-induction. After 48 h, cells were pelleted at 2000×*g* and resuspended in breaking buffer (50 mM sodium phosphate, 5% glycerol, 1 mM EDTA, pH 7.4) and frozen prior to processing.

Cell suspensions were thawed and lysed using a CF-1 cell disruptor at ~275 MPa chilled to 4 °C following the addition of 0.1% Triton-X 100. The resulting lysate was clarified through multiple rounds of centrifugation and a chemical precipitation step as previously described[16]. The clarified supernatants were then concentrated through a 30% sucrose cushion. The resulting pellet was resuspended in PBS + 1% NP40 + 0.5% sodium deoxycholate and clarified by centrifugation at 10,000×*g*. The supernatants were then purified through 15–45% sucrose gradients. Gradients were collected in 1 ml fractions from top to bottom and analysed for the presence of VLPs through immunoblotting and ELISA.

Briefly, for mammalian-expressed PV VLPs, PV-specific gene sequences for the P1 region were codon optimised for expression in mammalian cells and cloned into modified vaccinia virus Ankara (MVA) transfer vectors upstream of an uncleavable 3CD sequence derived from PV1 Mahoney with native sequence. In the resulting

dicistronic cassettes co-expression of 3CD alongside P1 was regulated using a PV3-derived internal ribosome entry site (PV-IRES). Additional elements of the expression cassette included an FMDV IRES upstream of P1, the FMDV 3′UTR downstream of 3CD followed by a 20 nucleotide long polyA tail, as well as the T7 promoter and terminator elements[15].

PV VLPs were produced by dually infecting MVA-PV recombinants harbouring a P1-3CD cassette with MVA-T7 viruses, expressing T7 polymerase in BHK-21 cells at 30 °C for 12 h. Cell suspensions were lysed by freeze-thawing, and PV VLPs were purified from clarified supernatants by concentration through a 30% sucrose cushion followed by a 15–45% sucrose gradient in 1× Dulbecco's phosphate-buffered saline (DPBS, Gibco), 20 mM EDTA, pH 7.0 (DPBS-EDTA)[15].

**Cryo-EM sample preparation and data collection.** Sucrose gradient purified fractions of the PV3-SC8 or wt PV2 VLPs expressed in yeast and mammalian cells, respectively, were pooled, buffer exchanged into DPBS-EDTA using Zeba Spin Desalting Columns (Thermo Fisher Scientific) with a 7 K molecular weight cut-off (MWCO) and concentrated using Amicon Ultra centrifugal filter devices (100 kDa MWCO, Merck Millipore) to a concentration of 0.9–1.5 mg/ml.

For the PV3-SC8$^{GPP3+GSH}$ and PV3-SC8$^{pleconaril+GSH}$ complexes, PV3-SC8 VLP and GPP3 or pleconaril were mixed at a ratio of 1 VLP:300 compound molecules, respectively, and incubated overnight at 4 °C, after which GSH was added to a final concentration of 10 mM and incubated on ice for 1–2 h. Compound 17 was ordered from a commercial supplier (www.wuxiapptec.com) and wt PV2 VLP and CP17 were mixed at a molar ratio of 1:2500 as used in the CVB3-CP17 and CVB4-CP48 studies[28,29] and incubated at 4 °C overnight. Both compounds were dissolved in DMSO at 10 mg/ml and diluted to appropriate working stocks as required. GSH stock solutions were prepared in distilled H$_2$O at pH 7.0.

Cryo-EM grid preparation was similar for all samples. Three to four microliters of VLP-compound or VLP-compound-GSH mixture were applied to glow-discharged Lacey carbon copper grids with an ultra-thin carbon support film (product No. AGS187-4, Agar Scientific). After 30 s, the unbound sample was removed by manual blotting with filter paper. To increase the number of particles in the holes, grids were re-incubated with a further 3–4 μl of sample for 30 s, followed by mechanical blotting for 3–4 s and rapid vitrification in a liquid ethane/propane mixture with a Vitrobot Mark IV plunge-freezing device (Thermo Fisher Scientific) operated at 4 °C and 100 % relative humidity.

For PV3-SC8$^{GPP3+GSH}$ and PV3-SC8$^{pleconaril+GSH}$, cryo-EM data acquisition was performed at 300 kV with a Titan Krios G3i microscope (Thermo Fisher Scientific) equipped with a Falcon III direct electron detector (DED) (Thermo Fisher Scientific) at the OPIC electron microscopy facility, UK. Micrographs were collected as movies using a defocus range of −2.9 to −0.8 μm in either single-electron counting mode (PV3-SC8$^{GPP3+GSH}$) or linear mode (PV3-SC8$^{pleconaril+GSH}$). Pixel sampling of 1.08 Å per pixel was used for both PV3-SC8$^{GPP3+GSH}$ and PV3-SC8$^{pleconaril+GSH}$ resulting in a calibrated magnification of ×129,629. Cryo-EM data for the wt PV2-CP17 complex were collected at 300 kV on a Titan Krios (Thermo Fisher Scientific) equipped with a K3 DED and GIF Quantum energy filter (Gatan) at the electron Bio-Imaging Centre, Diamond Light Source, UK. A total of 4750 micrograph images were collected as movies (50 frames, total electron exposure 34.68 $e^{-}$/Å$^2$) using a defocus range of −2.3 to −0.8 μm in single-electron counting mode at ×60,314 magnification, corresponding to a calibrated pixel size of 0.829 Å per pixel. Data were collected using EPU EM software (Thermo Fisher Scientific) for all samples. Data acquisition parameters are summarised in Table 1.

**Cryo-EM image processing.** For all datasets, image processing and single-particle reconstruction was performed using RELION-3.1[38] unless indicated otherwise. Individual movie frames were aligned and averaged with dose weighting using MotionCor2[39] to produce images compensated for electron beam-induced specimen drift. Contrast transfer function (CTF) parameters were estimated using CTFFIND4[40]. Micrographs showing astigmatism, significant drift or crystalline ice rings were discarded. For PV3-SC8$^{GPP3+GSH}$ and PV3-SC8$^{pleconaril+GSH}$ particle picking was performed in an automated manner with crYOLO[41] using a neural network trained model generated from the apo PV3-SC8 dataset[33], after which particle coordinates were saved and imported into RELION. For the wt PV2-CP17 data, particle picking was performed using programme ETHAN[42] within the Scipion software framework[43], after which saved particle coordinates were imported into RELION.

Single-particle structure determination used established protocols in RELION for image classification and gold-standard refinement to prevent over-fitting[44]. For all datasets, particles (numbers given in Table 1) were subjected to multiple rounds (at least two) of reference-free two-dimensional classification to discard bad particles and remove junk classes. The particle population for each dataset was further enriched by three-dimensional (3D) classification to remove broken and overlapping particles. The starting reference models for each dataset were either the previously determined cryo-EM structure of PV3-SC8 from a plant expression system[18] (EMDB accession code EMD-3747) low-pass filtered to 60 Å to avoid bias for PV3-SC8$^{GPP3+GSH}$ and PV3-SC8$^{pleconaril+GSH}$ or for wt PV2-CP17 a 50 Å low-pass filtered model was generated from a prior reconstruction of a subset of the data.

A final set of particles (numbers given in Table 1) were selected from the best-aligned 3D class averages for high-resolution 3D auto-refinement with the application of icosahedral symmetry throughout. For all datasets, a representative class from the end of 3D classification was low-pass filtered to 40 Å to avoid bias and used as a reference during refinement. After the first round of refinement, the datasets were subjected to CTF refinement to estimate beam tilt, anisotropic magnification, per-particle defocus and astigmatism, and also Bayesian polishing of beam-induced motion-correction with either default (wt PV2-CP17) or trained (PV3-SC8$^{GPP3+GSH}$ and PV3-SC8$^{pleconaril+GSH}$) parameters[38]. This procedure was performed iteratively either twice (PV3-SC8$^{GPP3+GSH}$ and wt PV2-CP17) or three times (PV3-SC8$^{pleconaril+GSH}$) with 3D auto-refinement after each round. The final resolution was estimated using a Fourier shell correlation (FSC) threshold of 0.143[32]. The maps for each reconstruction were sharpened using Post-processing in RELION by applying inverse $B$-factors of −52.3, −72.2 and −53.6 Å$^2$ for PV3-SC8$^{GPP3+GSH}$, PV3-SC8$^{pleconaril+GSH}$ and wt PV2-CP17, respectively. The wt PV2-CP17 data were subjected to Ewald sphere correction[35] to further improve the resolution, followed by sharpening with an ad-hoc $B$-factor of −15.0 Å$^2$ after testing a range of values for map interpretability. Local resolution was estimated for each reconstruction using the RELION implementation of local resolution algorithm[35], and locally scaled maps were used for model building and refinement in all cases. Data processing statistics are summarised in Table 1.

**Atomic model building, refinement and analysis.** For PV3-SC8$^{GPP3+GSH}$ and PV3-SC8$^{pleconaril+GSH}$, the atomic coordinates of the previously determined structure of PV3-SC8 (PDB ID 5O5B) were manually placed into the cryo-EM electron potential maps using UCSF Chimera[45]. Manual fitting was optimised with the UCSF Chimera 'Fit in Map' command[45] and the 'Rigid Body Fit Molecule' function in Coot[46]. For the wt PV2-CP17 structure, the atomic coordinates of PV2 Lansing strain (PDB ID 1EAH)[47] were used and a similar procedure was applied to optimise the initial fit. For all structures, the cryo-EM map surrounding a single capsid protomer (subunits VP0, VP1 and VP3) was extracted using phenix.map_box within Phenix[48]. Manual rebuilding was performed on these models using the tools in Coot[46], followed by iterative positional and B-factor refinement in real-space using phenix.real_space_refine[49] within Phenix[48]. All refinement steps were performed in the presence of hydrogen atoms. Chemical restraints for GPP3, pleconaril and CP17 were generated using the grade server[50]. Only atomic coordinates were refined; the maps were kept constant. Each round of model optimisation was guided by cross-correlation between the map and the model. Final models were validated using MolProbity[51], EMRinger[52] and CaBLAM[53] integrated within Phenix[48]. Refinement statistics are shown in Table 2.

Interface analysis of the PV3-SC8$^{GPP3+GSH}$, PV3-SC8$^{pleconaril+GSH}$ and wt PV2-CP17 binding pockets was performed using the 'Protein interfaces, surfaces and assemblies' service PISA at the European Bioinformatics Institute. (http://www.ebi.ac.uk/pdbe/prot_int/pistart.html)[54]. Gap-penalty-weighted structural superpositions of capsid protomers were performed with a version of the programme SHP[55] modified to estimate the evolutionary distance[56,57]. A full matrix of evolutionary distances was calculated, and the phylogenetic tree representation was generated from this distance matrix, using the programmes FITCH and DRAWTREE, as part of the PHYLIP package[58]. Sequence alignments were generated with Clustal Omega[59] and Espript (https://espript.ibcp.fr)[60]. Ligand interaction diagrams were prepared using Ligplot+ v.2.2[61]. Molecular graphics were generated using Pymol[62] and UCSF ChimeraX[63].

**Stability measurements of GSH binding to PV VLP.** The temperature at which a conformational change from D to C antigenicity occurred in the presence and absence of GSH was determined by heating at a range of temperatures from 30–70 °C followed by D antigen ELISA. Samples were diluted in 6-salt PBS with or without GSH to twice the concentration required to obtain an OD of 1.0 in D antigen ELISA, duplicate samples heated for 10 min at each temperature were then diluted 1:1 with 2% dried milk in 6-salt PBS and cooled on ice. D antigen content was measured by a non-competitive sandwich ELISA assay developed to measure the D antigen content of poliovirus vaccines[14]. Briefly, twofold dilutions of antigen were captured with a serotype-specific polyclonal antibody, then detected using serotype-specific, D antigen-specific monoclonal antibodies followed by anti-mouse peroxidase conjugate. The monoclonal antibodies used were 234 for type 1, 1050 for type 2 and 520 for type 3.

**Reporting summary.** Further information on research design is available in the Nature Portfolio Reporting Summary linked to this article.

## Data availability
The atomic coordinates for PV3-SC8$^{GPP3+GSH}$, PV3-SC8$^{pleconaril+GSH}$ and wt PV2-CP17 have been submitted to the Protein Data Bank under accession codes 8AYX, 8AYY and 8AYZ, respectively. The cryo-EM electron potential maps for PV3-SC8$^{GPP3+GSH}$, PV3-SC8$^{pleconaril+GSH}$ and wt PV2-CP17 have been deposited in the Electron Microscopy Data Bank under accession codes EMD-15725, EMD-15726 and

EMD-15727, respectively. The source data underlying Fig. 4 are provided in Supplementary Data 1. The datasets generated and/or analysed during the current study are available from the corresponding authors upon reasonable request.

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

## Acknowledgements

Computation was performed at the Oxford Biomedical Research Computing (BMRC) facility, a joint development between the Wellcome Centre for Human Genetics and the Big Data Institute supported by Health Data Research UK and the NIHR Oxford Biomedical Research Centre. Financial support was provided by the Wellcome Trust Core Award Grant Number 203141/Z/16/Z. Electron microscopy provision was provided through the OPIC electron microscopy facility, a UK Instruct-ERIC Centre, which was founded by a Wellcome JIF award (060208/Z/00/Z) and is supported by a Wellcome equipment grant (093305/Z/10/Z). We are grateful for technical assistance from the OPIC staff. We acknowledge Diamond Light Source for access and support of the cryo-EM facilities at the UK's national Electron Bio-imaging Centre (eBIC), instrument eBIC Titan Krios IV [under proposal EM20223-37], funded by the Wellcome Trust, MRC and BBRSC. We are grateful for technical assistance from the Diamond staff. M.W.B. and H.F. are supported by a WHO/Gates foundation award (RG.IMCB.I8-TSA-083) and C.P. by the Gates Foundation (OPP1192002). V.N. was supported by an EMBO Short-Term Fellowship. D.I.S. and E.E.F. are supported by the UK Medical Research Council (MR/N00065X/1) and E.E.F by the Wellcome Trust (101122/Z/13/Z). L.S., K.G., N.J.S. and D.J.R were funded by WHO 2019/883397-O "Generation of virus free polio vaccine—phase IV". This work was performed as part of a collaborative effort funded by the WHO/Bill and Melinda Gates Foundation (RG.IMCB.I8-TSA-083) involving the following Institutions and individuals: University of Leeds: D.J. Rowlands, N.J. Stonehouse, L. Sherry, K. Grehan. University of Oxford: D.I. Stuart, E.E. Fry, M.W. Bahar, C. Porta. John Innes Institute: G. Lomonosoff, D. Ponndorf. University of Florida: J.B. Flanegan, J. Morasco. National Institute for Biological Standards and Control: A.J. Macadam, P. Minor, H. Fox, S. Carlyle. The authors wish to thank Jim Hogle, Ellie Ehrenfeld, Philip Minor and Jeff Almond for their support and invaluable scientific input.

## Author contributions

Experiments were conceived and designed by M.W.B. and D.I.S. M.W.B., V.N., H.F., L.S. and K.G. performed experiments. M.W.B. and V.N. collected and processed the cryo-EM data, built and refined atomic models and, along with C.P., E.E.F. and D.I.S. analysed cryo-EM results. H.F. performed the GSH thermostability assay and, along with A.J.M. analysed the results. All authors, including N.J.S. and D.J.R. interpreted the results. M.W.B., C.P., H.F., E.E.F. and D.I.S. wrote the manuscript, and all authors reviewed and edited the manuscript.

## Competing interests

The authors declare no competing interests.
