## [Peer Review File · Communications Biology]

Reviewers' comments:

Reviewer #1 (Remarks to the Author):

A conserved glutathione binding site in poliovirus is a target for antivirals and vaccine stabilization by Bahar et al

In this manuscript, the authors investigate the stabilizing effect of GSH on poliovirus VLPs. They suggest that this is important for a better understanding of strategies to improved immunogenicity of VLPs. They employ mainly single particle cryoEM for a structural characterization of the factor binding pockets.

Overall, the manuscript is well written and I have only minor comments and suggest acceptance following this review.

1. The authors suggest that the rather abundant cellular GSH will function as a stabilizing agent for VLPs. If so, one would expect some evidence of GSH to co-purify with VLPs. Furthermore, the well-coordinated GSH binding site detailed here would argue a rather high occupancy of GSH in the VLPs. Yet, the apo state cryoEM structure does not seem to suggest any feature corresponding to natively bound GSH. This is seemingly a contradiction from the postulated role of GSH.
2. The premise of the study is that GSH is critical for VLP stability and thus, for immunogenicity. How do the authors envision providing GSH in the VLPs? In bulk in vaccine stocks? Are there examples of this approach? Otherwise, the context in which the case is made to study these VLPs seems rather weak.
3. Minor point: The authors go back and forth between "potential" and "density" in describing the cryoEM map. I suggest adhering to potential, which is correct in strict physics-related terms. One could use "feature" to replace density.
4. It is not surprising that Ewald sphere correction did not yield any substantial improvements ($\sim 0.1\text{\AA}$) since the particles are on the smaller side. This is relative to say, giant viruses and even alphaviruses. It seems almost superfluous to spend real estate on this Ewald correction.

Appropriateness and validity of any statistical analysis: Acceptable

Reviewer #2 (Remarks to the Author):

The recent discovery of circulating poliovirus in London and New York is highlighting the importance of the introduction of better vaccines for poliovirus elimination. The manuscript by Bahar and colleagues brings important structural information to a central problem regarding the construction of future the polio vaccines: the design of a stable virus like particle (VLP). The authors describe the structural basis of poliovirus VLP stabilization through the binding of a glutathione (GSH) molecule in a positively charged pocket located at the VP1-VP3 interprotomer interface.

Importantly, the structural analysis reveals that GSH binding pocket is highly conserved in enteroviruses, supporting the existence of a common assembly mechanisms for the majority of enteroviruses. Such mechanistic details are essential for the successful development of therapies, such as viral inhibitors and vaccines.

The study is of very high quality. For instance, the manuscript reports one of the highest resolution maps of a viral capsid and the highest picornavirus structure deposited to date. The text is clear and fluent; the figures are informative with detailed legends.

Overall, I consider that the manuscript would be of high interest for structural biologists and researchers involved in optimizing vaccine construction.

Reviewer #3 (Remarks to the Author):

Comments:

The manuscript titled " A conserved glutathione binding site in poliovirus is a target for antivirals

and vaccine stabilisation" by Mohammad et al constitutes a comprehensive study on high-resolution cryo-EM structures of poliovirus in complex with GSH, which reveals a conserved binding site in many enteroviruses. Poliovirus, an important class of human pathogens, belonging to the genus of Enterovirus within the family of Picornaviridae, can cause poliomyelitis. In addition, polioviruses also serve as models for understanding the basic mechanisms of host-pathogen interactions, virus entry and viral genome release. It is of importance to characterize the structural details to facilitate stabilized vaccine and small-molecule drug design. The authors describe a number of high-resolution cryo-EM structures of PV in complex with GPP3/GSH/CP17/ pleconaril or combinations of these compounds and demonstrated that GSH could stabilize substantially viral particles. These information will guide rational drug design against many enteroviruses. This manuscript is clearly written and the figures are good. Data analysis seems technically valid. It deserves publication in Communications Biology. However, a number of concerns need to be addressed before the formal publication.

Specific points:

1. Line 143-143, The overall structure of the GSH-bound PV3-SC8 VLP is essentially identical to the unbound native antigenic form of PV3-SC8... with r.m.s.d of 0.7, 1.4 and 1.5 Å... The overall r.m.s.d of 1.5 Å suggests subtle structural difference. Please clarify the notable difference in specific regions with r.m.s.d of over 2 Å, presumably somewhere around the binding sites.

2. Lines 149-152, so effects at the two druggable sites, separated by some 22 Å in the capsid are likely to be additive. Structurally, it's reasonable. Is it possible to provide experimental evidence for the synergic effects? Or comment this point in the discussion section.

Minor points:

1. The color bar (grey for neighbouring protomer) should be provided in the Figure 1d.

2. Fig.3a, sequence alignments from residues 108-114 in VP1 seem not be involved in GSH binding and were not discussed in the main text. Please remove this region. In addition, a few residues labelled by pink spheres were not mentioned in the main text.

3. Fig.4, It's difficult to distinguish these small labels. Lines shown in different colors are suggested.

4. The authors are suggested to discuss if GSH can bind expanded particles, the 'C' antigenic conformation, for example.

We have modified the manuscript as suggested.
Point-by-point rebuttal:

Reviewer #1 (Remarks to the Author):

A conserved glutathione binding site in poliovirus is a target for antivirals and vaccine stabilization by Bahar et al

In this manuscript, the authors investigate the stabilizing effect of GSH on poliovirus VLPs. They suggest that this is important for a better understanding of strategies to improved immunogenicity of VLPs. They employ mainly single particle cryoEM for a structural characterization of the factor binding pockets.

Overall, the manuscript is well written and I have only minor comments and suggest acceptance following this review.

1. The authors suggest that the rather abundant cellular GSH will function as a stabilizing agent for VLPs. If so, one would expect some evidence of GSH to co-purify with VLPs. Furthermore, the well-coordinated GSH binding site detailed here would argue a rather high occupancy of GSH in the VLPs. Yet, the apo state cryoEM structure does not seem to suggest any feature corresponding to natively bound GSH. This is seemingly a contradiction from the postulated role of GSH.

This is a reasonable question. We know that GSH is required for assembly of the virus and we presume that it will be bound to the VLPs when they are intracellular. It seems most likely that the off-rate is sufficiently fast that when extensively purified the GSH is stripped off (in addition the level of overexpression may deplete cellular GSH). The stability of the virus particle is a balance with disassembly required for infection and overstabilisation (as occurs with pocket binding drugs) would block infection. The results are therefore not surprising. It is possible that, since VLPs could be overstabilised, modification of the virion to further enhance affinity might be a useful avenue to pursue in vaccine antigen design, however that is beyond the scope of the present paper. We have modified the text to address this point.

2. The premise of the study is that GSH is critical for VLP stability and thus, for immunogenicity. How do the authors envision providing GSH in the VLPs? In bulk in vaccine stocks? Are there examples of this approach? Otherwise, the context in which the case is made to study these VLPs seems rather weak.

Yes, the expectation is that GSH would be used as an excipient added to the stocks.

3. Minor point: The authors go back and forth between "potential" and "density" in describing the cryoEM map. I suggest adhering to potential, which is correct in strict physics-related terms. One could use "feature" to replace density.

Potential/density – we have followed the advice of the referee.

4. It is not surprising that Ewald sphere correction did not yield any substantial improvements (~0.1Å) since the particles are on the smaller side. This is relative to say, giant viruses and even alphaviruses. It seems almost superfluous to spend real estate on this Ewald correction.

Ewald sphere - agreed, it would only strongly kick in at somewhat higher resolution for particles of this size. We have reduced the Ewald real estate in the main paper but retained in supplementary, since although small it was a significant improvement.

Appropriateness and validity of any statistical analysis: Acceptable

Reviewer #2 (Remarks to the Author):

The recent discovery of circulating poliovirus in London and New York is highlighting the importance of the introduction of better vaccines for poliovirus elimination.

The manuscript by Bahar and colleagues brings important structural information to a central problem regarding the construction of future the polio vaccines: the design of a stable virus like particle (VLP). The authors describe the structural basis of poliovirus VLP stabilization through the binding of a glutathione (GSH) molecule in a positively charged pocket located at the VP1-VP3 interprotomer interface.

Importantly, the structural analysis reveals that GSH binding pocket is highly conserved in enteroviruses, supporting the existence of a common assembly mechanisms for the majority of enteroviruses. Such mechanistic details are essential for the successful development of therapies, such as viral inhibitors and vaccines.

The study is of very high quality. For instance, the manuscript reports one of the highest resolution maps of a viral capsid and the highest picornavirus structure deposited to date. The text is clear and fluent; the figures are informative with detailed legends.

Overall, I consider that the manuscript would be of high interest for structural biologists and researchers involved in optimizing vaccine construction.

Reviewer #3 (Remarks to the Author):

Comments:

The manuscript titled "A conserved glutathione binding site in poliovirus is a target for antivirals and vaccine stabilisation" by Mohammad et al constitutes a comprehensive study on high-resolution cryo-EM structures of poliovirus in complex with GSH, which reveals a conserved binding site in many enteroviruses. Poliovirus, an important class of human pathogens, belonging to the genus of Enterovirus within the family of Picornaviridae, can cause poliomyelitis. In addition, polioviruses also serve as models for understanding the basic mechanisms of host-pathogen interactions, virus entry and viral

genome release. It is of importance to characterize the structural details to facilitate stabilized vaccine and small-molecule drug design. The authors describe a number of high-resolution cryo-EM structures of PV in complex with GPP3/GSH/CP17/pleconaril or combinations of these compounds and demonstrated that GSH could stabilize substantially viral particles. These information will guide rational drug design against many enteroviruses. This manuscript is clearly written and the figures are good. Data analysis seems technically valid. It deserves publication in Communications Biology. However, a number of concerns need to be addressed before the formal publication.

Specific points:

1. Line 143-143, The overall structure of the GSH-bound PV3-SC8 VLP is essentially identical to the unbound native antigenic form of PV3-SC8.... with r.m.s.d of 0.7, 1.4 and 1.5 Å... The overall r.m.s.d of 1.5 Å suggests subtle structural difference. Please clarify the notable difference in specific regions with r.m.s.d of over 2 Å, presumably somewhere around the binding sites.

We had been concerned about this and went back and carefully recalibrated the microscope magnification (using the full particle as a measure). Having corrected this the RMSD of 1.5 Å is reduced to 0.77 Å. The revised manuscript has been updated.

2. Lines 149-152, so effects at the two druggable sites, separated by some 22 Å in the capsid are likely to be additive. Structurally, it's reasonable. Is it possible to provide experimental evidence for the synergic effects? Or comment this point in the discussion section.

Unfortunately we have no evidence for this, and it would be very hard for us to perform further experiments. For the purposes of vaccine production the current generation of stabilised VLPs will be stable enough without an additive effect. We have slightly modified the discussion as suggested.

Minor points:

1. The color bar (grey for neighbouring protomer) should be provided in the Figure 1d.

Done.

2. Fig.3a, sequence alignments from residues 108-114 in VP1 seem not be involved in GSH binding and were not discussed in the main text. Please remove this region. In addition, a few residues labelled by pink spheres were not mentioned in the main text.

Thank you, we have corrected this. Residue 113 in VP1 (blue sphere) and residues 233-236 (pink spheres) are involved in forming the GSH binding site and Fig. 3a updated accordingly. These are also labelled on Fig. 2a and 2b and specific residues labelled by pink spheres that form hydrophobic interactions with GSH are referred to in the text.

3. Fig.4, It's difficult to distinguish these small labels. Lines shown in different colors are suggested.

Thanks, done.

4. The authors are suggested to discuss if GSH can bind expanded particles, the 'C' antigenic conformation, for example.

We have expanded consideration of this important point in the discussion and included a supplementary figure 4 to demonstrate this point.

REVIEWERS' COMMENTS:

Reviewer #3 (Remarks to the Author):

The authors have addressed all my concerns and the revised manuscript has been significantly strengthened. I suggest a quick publication.